# Vimentin Dynamics in Viral Infection: Shield or Sabotage?

**DOI:** 10.3390/ijms27010388

**Published:** 2025-12-30

**Authors:** Ying Ling, Xuanyi Ling, Zaixin Liu

**Affiliations:** 1School of Biosciences, University of Birmingham, Birmingham B15 2TT, UK; 2Institute of Biophysics, Chinese Academy of Sciences, Beijing 100101, China; 3Academic Center for Exploratory Students, University of Connecticut, Storrs, CT 06269, USA; ugi24002@uconn.edu; 4State Key Laboratory of Veterinary Etiological Biology, Lanzhou Veterinary Research Institute, Chinese Academy of Agricultural Sciences, Lanzhou 730046, China; 5National Foot and Mouth Disease Reference Laboratory, Lanzhou Veterinary Research Institute, Chinese Academy of Agricultural Sciences, Lanzhou 730046, China

**Keywords:** vimentin, virus, co-receptor, PTMs, vimentin cage, viral replication complexes

## Abstract

Vimentin is a type III intermediate filament protein that maintains cellular integrity, organelle positioning, and resilience to mechanical stress, but it is increasingly recognized for its dynamic change in viral infection. Viral infection causes vimentin filament disassembly into soluble oligomers with hydrophobic and acidic interfaces conducive to viral binding. These oligomers are recruited to the cell surface, where they act as viral co-receptors, facilitating viral attachment and entry. Upon entry, the viral protein induces post-translational modifications in intracellular vimentin filaments undergoing rearrangement processes, including disassembly into oligomers and then reassembly into cage-like structures that encapsulate viral replication complexes. Whether these structures promote viral replication or represent a host-imposed defense remains open. Our findings highlight the pro-viral “shield” and anti-viral “sabotage” role, a context-dependent role of vimentin during viral infection. Importantly, we offer a perspective encompassing structural biology and molecular and cellular signaling insights into vimentin dynamics, an approach that has not been explored in the current literature. We further propose that targeting vimentin is an innovative strategy for anti-viral intervention.

## 1. Introduction

Intermediate filaments (IFs) are a family of tissue- and context-specific cytoskeletal proteins [1,2,3]. Vimentin is a type III IF protein predominantly expressed in mesenchymal cells [4]. Structurally, vimentin comprises an intrinsically disordered N-terminal head domain, a central α-helical rod domain responsible for coiled-coil dimer formation, and a flexible C-terminal tail domain that contributes to filament assembly and interaction specificity [5]. These domains enable vimentin to assemble into dynamic hierarchically organized filament networks that sustain cellular plasticity, anchor organelles, and provide mechanical stability. Beyond its structural role, vimentin participates in a wide range of cellular processes through signaling pathways and gene regulatory networks, including wound healing, cellular remodeling, and tissue repair [6]. Vimentin is increasingly recognized as a central mediator of cell activation, with emerging roles in inflammation, apoptosis, and cellular signaling [7,8,9]. Vimentin exists in both extracellular and intracellular forms. Furthermore, extracellular vimentin can be divided into cell surface vimentin (CSV), which remains anchored to the cell membrane, and secreted vimentin, which is released into the extracellular environment. CSV represents a dynamic form of vimentin that is re-localized from the intracellular cytoskeleton to the plasma membrane or the extracellular environment [10]. Unlike its classical role as a cytoplasmic structural component, CSV performs non-structural functions, particularly in pathogen recognition, receptor organization, and efficient cell-to-cell spread.

The intracellular filaments of vimentin are dynamic structures that can disassemble and reassemble in response to specific post-translational modifications (PTMs) like phosphorylation, proteolysis, or oxidation [11,12]. Specifically, phosphorylation mediated by virus-activated host signaling pathways, such as protein kinase C (PKC) [13], Rho-kinase, and calcium/calmodulin-dependent protein kinase II (CaMKII), promotes the disassembly of vimentin filaments, and the dephosphorylated oligomers undergo a second phase of conformational rearrangements, converting vimentin from an oligomeric to a polymeric state [14]. One notable consequence of this structural remodeling is the formation of vimentin-based cage-like structures that encapsulate virus replication complexes (VRCs), which are often at pericentriolar sites [15]. Whether these pericentriolar assembly sites represent a host defense mechanism or virus-driven strategy, and whether all viruses exploit this vimentin-based architecture, remains unclear. In specific infections, these structures provide both physical protection and spatial organization to the viral replication machinery, thereby increasing its efficiency. However, vimentin plays a dual role during viral replication—it can either support viral replication by promoting cage formation or impede it by disrupting viral trafficking and enhancing apoptosis [16,17,18].

Here, this review summarizes the pro-viral “shield” and anti-viral “sabotage” roles of dynamic vimentin during viral infection, influencing viral entry and replication. Accumulating evidence indicates that the effects of vimentin are context-dependent, differing among different virus families or host cell signaling environments. Elucidating the molecular basis underlying these diverse outcomes will expand our understanding of vimentin biology and provide a theoretical framework for evaluating its potential as a target for anti-viral intervention.

## 2. Structure and Function of Vimentin

### 2.1. Structural Assembly Pathway of Vimentin

Vimentin is a 54 kDa polypeptide composed of 466 amino acids. AlphaFold-predicted structural models indicate that human vimentin is a mixed α/β protein, featuring a highly conserved α-helical “rod” domain flanked by a non-α-helical N-terminal “head” domain and a two-stranded antiparallel β-sheet-rich C-terminal “tail” domain; the linker domain is interspersed between distinct sub-helices within the rod domain (Figure 1). Vimentin dimers associate via homodimerization in a half-staggered antiparallel manner to form soluble tetramers, which are the basic cytoplasmic vimentin subunits in vivo [19].

A hallmark of vimentin is its highly dynamic structural behavior and capacity to form a cage-like filamentous network around the nucleus [20]. Assembly begins with the nucleation of soluble oligomers in the perinuclear region, where monomers associate to form parallel coiled-coil dimers through interactions along the central 1A–2B rod domains. The two dimers align in an antiparallel fashion to form A11- and A22-type tetramers, with overlaps at the N-terminal coil 1B–2A and C-terminal 2B–2B regions, respectively. In contrast, the antiparallel C–N (ACN) tetramer is formed by coupling the C-terminal region of one dimer with the N-terminal region of another [21,22]. These tetramers further elongate into protofibrils via longitudinal and lateral interactions. Then, five octameric protofibrils laterally associate and are connected by tail domain interactions, whereas the head domains cluster to form a central luminal fiber, which is subsequently integrated into the peripheral filament network [23]. As IFs mature, they bundle into a dense perinuclear cage-like network that supports nuclear positioning, preserves nuclear integrity, and enhances cellular resilience under stress. They also stabilize the positioning of organelles, including the mitochondria and nucleus, thereby preventing rupture. This protective role is further reinforced through scaffolding and sequestration as filaments form cage-like structures around protein aggregates or pathogens, thus isolating them from the cytoplasm [24]. This raises the critical question of whether viruses repurpose the naturally protective vimentin cage as a pro-viral platform, effectively converting a host protective structure into a tool for shielding viral replication.

### 2.2. Characteristics of the Vimentin Oligomers

Structural data from cryo-electron microscopy (cryo-EM) and electron tomography have revealed three-dimensional models of truncated monomeric and oligomeric states of vimentin, which are now available in the Protein Data Bank (PDB) [25]. Although the crystal structure of the full-length protein is not available, structurally homologous truncated constructs ranging from monomers (PDB: 4YPC; 161–243 aa) to tetramers (PDB: 3KLT; 253–324 aa), hexamers (PDB: 1GK4; 328–411 aa), and octamers (PDB: 5WHF; 153–238 aa) have been characterized at multiple assembly stages [26]. Notably, the recently deposited structure of the assembled vimentin IFs (PDB: 8RVE; 1–466 aa) provides compelling evidence for biopolymer assembly with five well-defined protofibrils [23].

To investigate how structural transitions affect membrane interactions, we analyzed the molecular lipophilicity potential (MLP) of vimentin across different assembly states using ChimeraX 1.9 [27]. MLP calculations project lipophilic and hydrophilic atomic contributions onto a three-dimensional molecular surface—a molecular property encoding intermolecular recognition and intramolecular interactions [28]. Monomeric and low-order oligomeric vimentin adopt relatively open conformations, exposing hydrophobic and hydrophilic patches (Figure 2a,b). This mixed surface polarity likely enhances its affinity for lipid membranes, suggesting that oligomeric vimentin is more prone to membrane-related functions. For viruses, such oligomeric conformations may be more accessible or responsive to viral manipulation. In contrast, polymerized vimentin forms compact filaments wherein hydrophobic residues are buried at the subunit interfaces, whereas hydrophilic residues (such as Arg, Lys, and Glu) are exposed on the surface (Figure 2c). Consequently, the membrane-facing regions of the mature filaments are predominantly hydrophilic, which substantially limits their capacity to interact with lipid biolayers compared with the more open and surface exposed oligomeric forms. Based on the observed differences in surface lipophilicity potential between oligomeric and filamentous vimentin, we propose the following testable hypothesis: viruses may preferentially exploit soluble or oligomeric vimentin as membrane anchors, whereas the mature filament network preferentially functions as a hydrophilic structural scaffold that localizes and stabilizes phase-separation process surrounding VRCs.

The Coulombic electrostatic potential values calculated using PyMOL 3.1.4.1 were used to investigate the electrostatic interactions and structural features of the different oligomeric states of vimentin. PyMOL has long been a widely adopted tool for structural bioinformatics and supporting applications such as molecular docking, structure-function relationship analysis, protein structure prediction, and virtual screening [29]. Here, we focus on tetramers 3KLT and 3UF1, wherein the dimers are arranged antiparallel to the mixed termini. 3KLT (containing the 2A–2B domain) displays an overall even electrostatic potential distribution. In the front view, there is a prominent negative potential in the middle region, with additional negative patches toward the C-terminus, while some positive and neutral potentials appear at the N-terminals, mainly at the top and bottom (Figure 3a, top view). In the side view, the 3KLT electrostatic surface displays seven negatively charged patches symmetrically separated around the central axis of the groove. Positively charged patches enriched in Arg and Lys are distributed near the outer edges, whereas negative and minor neutral potentials occupy the central region (Figure 3a, bottom view). 3UF1, which represents the vimentin–1B A11 homotetramer, contains an N-terminal hydrophobic pocket and a C-terminal anchoring knob with a generally conserved tetramer assembly mechanism [22]. One homodimer displays a long, continuous groove characterized by a predominance of negative charges in the front view of the electrostatic surface, whereas positively charged patches are sparse and isolated, either within the groove or near the molecular edges (Figure 3b, top view). In the side view (rotated 90°), the neutral and positive potentials are more evenly distributed across the central surface, whereas strong negative potentials persist along the sides. Consequently, we propose a second testable hypothesis: the regularity and surface exposure of these positive areas make both a promising vimentin interaction module for use in viruses through electrostatic complementarity and groove-mediated binding.

### 2.3. PTMs Trigger Vimentin Remodeling

Vimentin remodeling enables the dynamic reorganization of the cytoskeleton, allowing cells to adapt to changing environmental stressors. In response to specific external cues, intracellular signaling cascades initiate the polymerization of vimentin monomers into robust IFs [20], which are essential for maintaining cell shape, ensuring membrane integrity, and organizing the overall cellular architecture. Conversely, specific signals can trigger the disassembly of IFs into smaller oligomeric units, such as tetramers or dimers, thus facilitating rapid cytoskeletal reconfiguration in response to acute stress. Although vimentin is encoded by a single gene, its functional diversity is greatly enhanced by numerous PTMs, including phosphorylation, proteolysis, and acetylation, which modulate its assembly dynamics, cellular localization, and protein-protein interactions [14].

Phosphorylation is a major PTM that regulates vimentin filament disassembly. According to the PhosphoNET database (http://www.phosphonet.ca/; accessed on 15 October 2025), vimentin has 74 reported phosphorylation sites, primarily targeting serine and threonine residues with fewer modifications to tyrosine residues. Phosphorylation is tightly regulated by several cell cycle-related kinases and signaling molecules, including CDK1, PKC, Rho-kinase, CaMKII, and the constitutively active Cdc42^V12^ mutant [13,30,31,32]. Phosphorylation introduces negative charges that alter the charge distribution and polarity of proteins, causing conformational rearrangements that disrupt filament assembly and stability [33]. This modification weakens the interactions between IFs, thereby promoting their disassembly and increasing their solubility [14]. Conversely, dephosphorylation promotes the assembly of soluble oligomeric intermediates into mature filaments. Phosphorylation and dephosphorylation transitions in vimentin organization may contribute to conditions that favor interactions with the viral protein [34]. Likewise, vimentin is regulated by proteolytic cleavage through enzymes such as calpain, caspases [35], and viral proteases [36,37,38], which cleave vimentin into smaller fragments. For instance, calpain-mediated cleavage facilitates the secretion of vimentin oligomers into the extracellular space, thereby contributing to cellular adaptive responses [39,40].

Vimentin behavior is modulated by other PTMs. Lys120 acetylation neutralizes the positive charge of lysine, reducing electrostatic interactions between vimentin molecules and promoting filament disassembly [41]. Ubiquitination attaches ubiquitin molecules to lysine residues, targets vimentin for degradation, and affects cellular function [42]. S-glutathionylation of the conserved Cys328 residue acts as a signaling mechanism during oxidative stress, disrupting the longitudinal assembly of unit-length filaments (ULFs) and compromising filament network stability [43]. Therefore, Cys328 is a major target for the oxidative and electrophilic modifications that affect vimentin structure and function [21]. In contrast to disassembly promoting PTMs, O-linked glycosylation favors vimentin assembly. This modification attaches N-acetylglucosamine to serine or threonine residues in the head domain, including Ser34, Ser39, and Ser49 [44]. Specifically, glycosylation at Ser49 promotes dimer assembly, stabilizes filaments, and regulates cell migration [45].

## 3. Cell Surface Vimentin as a Pro-Viral “Shield” Role Facilitating Viral Entry

Recently, an increasing number of studies have reported that viral attachment and entry often rely on the externalization of vimentin, a process by which intracellular vimentin is translocated to the cell surface and predominantly exists as oligomers comprising approximately 4–12 monomers, which are smaller than a single ULF (eight tetramers) [46]. This oligomeric form facilitates viral attachment, internalization, and subsequent infection. Multiple viral families, including coronaviruses, flaviviruses, picornaviruses, herpesviruses, and hepaciviruses, exploit CSV as a co-receptor or attachment factor to enhance binding efficiency and entry, underscoring its pro-viral “shield” role during infection initiation (Table 1).

### 3.1. Coronavirus

Accumulating evidence indicates that CSV functions not merely as an auxiliary attachment factor but as an active structural co-receptor that modulates the spatial organization of coronavirus entry. CSV as a binding partner bridging host angiotensin-converting enzyme 2 (ACE2) and the receptor-binding domain (RBD) of the viral spike (S) protein [57,58]. Early studies suggested that targeting vimentin by reducing CSV expression could be a promising therapeutic approach to limit *SARS-CoV-2* infection [52,59]. However, recent biophysical measurements have begun to clarify how the structural state of CSV enables this function. Using single-molecular force spectroscopy, Deptuła et al. [47] demonstrated that native extracellular vimentin—present predominantly as non-filamentous oligomers rather than mature filaments—exhibits markedly higher binding forces toward the S protein compared to vimentin-deficient controls. This is consistent with other studies reporting that vimentin oligomers exist on the surface of the cell and bind to the S protein of *SARS-CoV-2* [47,48,49].

Integrating these experimental observations with computational modeling, Deptuła et al. [47] further show that the C-terminal region of vimentin (390–466 aa), comprising the coil 2B segment of the rod domain, likely forms the primary docking interface with the S protein. This is in agreement with findings from Lam et al. and Suprewicz et al. [49,50], who demonstrated that the recombinant rod domain—especially the coiled-coil dimeric (261–335 aa) form—established more extensive hydrogen bonding and electrostatic networks with the S protein than the monomeric fragment (102–138 aa). In addition to these structural insights, wet-lab studies have also demonstrated that both the recombinant rod domain of vimentin and vimentin-targeting agents [53,60] can effectively reduce *SARS-CoV-2* attachment by competitively blocking the CSV-S-ACE2 attachment axis in vitro. Importantly, these inhibitory effects arise from perturbation of vimentin-mediated interactions rather than from the absence of vimentin function itself. Taken together, these findings point toward a convergent mechanism in which the rod-domain dimer of CSV functions as a key component of the co-receptor complex for the initial S-protein attachment, thereby orienting the virion in a configuration that facilitates viral entry.

Notably, these mechanistic models also provide a unifying rationale for the anti-viral effects of vimentin-targeting molecules and recombinant rod-domain constructs, which likely function by disrupting this geometrically optimized CSV-S-ACE2 triad. However, most current models rely heavily on docking simulations or in vitro assays using recombinant fragments, leaving open important questions regarding the native structure heterogeneity and dynamic mobility of CSV on living cell membranes. Addressing these gaps will be crucial for validating CSV as a robust therapeutic target and for understanding how extracellular vimentin reorganizes to act as a pro-viral role supporting viral entry under physiological conditions.

### 3.2. Flaviviruses

Vimentin also acts as a co-receptor or attachment factor, facilitating flavivirus binding and internalization. *Japanese encephalitis virus (JEV)* is an enveloped single-stranded positive RNA virus; CSV interacts with its viral envelope glycoprotein and functions as a key binding molecule for JEV entry [51,61,62]. Viral binding was significantly inhibited by anti-vimentin antibodies and recombinant vimentin, and it was reduced in vimentin-knockdown cells. Notably, comparative analysis of two viral strains revealed that the envelope (E) protein of the virulent *JEV* RP-9 strain specifically interacts with the head and tail domains of vimentin. The point mutation E-E138K reduces this dependence [51], indicating that viral-vimentin interactions are finely tuned by strain-specific structural features of the E protein. This interaction likely involves electrostatic attractions between the positively charged E protein [63] and negatively charged vimentin oligomers, facilitating the concentration of virions on the host cell surface and promoting their interaction with endocytic receptors present on the surface of the host cell (e.g., α_v_β_3_ integrin). Another study showed that JEV activates the dopamine D2 receptor, which stimulates the phospholipase C (PLC) signaling pathway. This cascade leads to increased CSV expression in human dopaminergic neuronal cells, rendering them more susceptible to viral entry [62].

Consistently, dengue virus *(DENV),* a mosquito-borne virus belonging to the genus *Flavivirus* in the family *Flaviviridae*, uses CSV for entry. Yang et al. [64] found that the E protein domain III (EDIII) of *DENV-2* directly interacts with the rod domain (102–410 aa) of vimentin on the surface of vascular endothelial cells (VECs). Computational modeling further identifies discrete interaction hotspots: five residues of *DENV-2* EDIII (D53, F85, E82, G30, and H29) and six residues of the vimentin rod domain (Y291, L380, E288, Y383, L284, and M391). These structured contacts support a model in which CSV acts as an initial docking platform that aligns the virion for interaction with endothelial receptors. However, polyclonal antibodies (PcAbs) targeting the vimentin rod domain fail to fully block *DENV-2* entry, suggesting that additional factors—such as β3 integrin [65]—may also contribute to form a multicomponent viral docking complex. Further investigations are warranted to clarify the mechanisms underlying the secretion of superficial vimentin in VECs and to define the precise interaction sites involved in virus attachment.

Although CSV has been implicated in the entry of various viruses (Table 1), the mechanism by which vimentin is recruited to the cell membrane remains poorly understood. We speculate that during flavivirus infection, vimentin is recruited to the cell surface via a mechanism involving β3 integrin and plectin in a similar fashion [66]. Furthermore, β3 integrin associates with the E protein of *JEV* [67] and *DENV-2* [65], facilitating their entry into host cells. This model is supported by several lines of evidence: vimentin is recruited to the cell surface through a coordinated mechanism involving β3 integrin and the cytolinker protein plectin during viral pathogen infection. β3 integrin connects intracellular signaling networks and the cytoskeleton via its cytoplasmic tail, which contains key tyrosine residues essential for vimentin recruitment [68,69]. Moreover, plectin serves as a molecular bridge, linking β3 integrin to vimentin at the cell periphery [70,71], thereby favoring viral access to surface binding sites. This presents a testable hypothesis for CSV recruitment and an important avenue for future research.

### 3.3. Hepaciviruses

The *hepatitis C virus* (*HCV*) belongs to the genus *Hepacivirus*, a member of the family *Flaviviridae*. A recent study identified a previously unreported vimentin-dependent HCV cell–cell transmission mode, beyond classical mechanisms such as virological synapses and tunneling nanotubes (TNTs) [54]. While vimentin is dispensable for cell-free viral entry, its deletion selectively disrupts direct cell–cell spread in differentiated hepatocytes, indicating that specialized requirement for CSV during intercellular dissemination. Mechanistically, the N-terminal head domain (1–95 aa) of CSV directly interacts with the *HCV* E1 protein, facilitating virion attachment to recipient cell membranes and promoting efficient intercellular transmission. Importantly, the expression of either full-length vimentin or just its N-terminal head domain rescues the defective cell–cell transmission in vimentin-knockout cells, revealing that the vimentin head domain alone contains the functional determinants needed for *HCV* intracellular spread. Antibody-mediated targeting of CSV significantly inhibits *HCV* intercellular transmission by blocking the E1-binding site, underscoring CSV as a potential therapeutic target for limiting viral dissemination.

Despite these advances, several uncertainties remain. Current evidence is primarily derived from in vitro models of differentiated hepatocytes, and whether the same CSV-dependent mechanism operates in the complex architecture and immunological environment of the liver in vivo is not yet clear. Moreover, the mechanism by which CSV is externalized and maintained on the hepatocyte surface during *HCV* infection remains unresolved. It is also plausible that additional, as-yet unidentified co-receptors cooperate with CSV to stabilize *HCV* transmission, especially given the virus’s ability to exploit multiple redundant pathways to ensure persistence. Finally, whether *HCV* manipulates vimentin secretion or membrane distribution dynamically during infection, as for *SARS-CoV-2* and other viruses [64,72,73], has not been systematically examined. Addressing these gaps will be essential for determining whether targeting CSV can meaningfully restrict *HCV* dissemination in physiological settings and for clarifying the broader relevance of this mechanism to other hepatotropic viruses.

### 3.4. Picornaviruses

*Human enterovirus 71 (EV71)*, a non-enveloped single-stranded RNA virus (*Enterovirus, Picornaviridae*), enters host cells in three stages: attachment, endocytosis, and uncoating [74]. Du et al. [73] demonstrated that *EV71* can attach to CSV through its VP1 protein, facilitating the establishment of EV71 infection. Conversely, viral replication—indicated by VP1 protein expression, viral RNA levels, and virus yield—was reduced by competition with exogenous vimentin or antibodies targeting surface vimentin. Furthermore, knockdown of surface vimentin expression in U251 cells leads to decreased viral binding. However, *EV71* infection was not completely blocked, even at high concentrations of anti-vimentin antibody or soluble vimentin, suggesting that vimentin functions as a co-receptor or cofactor rather than as the sole entry receptor. The incomplete neutralization achieved by anti-vimentin strategies also suggests the presence of parallel entry pathways or compensatory receptors, which need to be mapped systematically. Supporting this concept, *EV71* binds vimentin and its primary receptor SCARB2 at distinct cell sites, with vimentin acting not as a competitor but as an attachment factor that facilitates infection by presenting the virus to its functional receptor [73]. At the structural level, the head domain (1–56 aa) of vimentin was found to be directly responsible for its specific binding to *EV71* VP1.

An important small breakthrough is the successful creation of the vimentin-deficient mouse model. Both in vitro and in vivo experiments revealed that the A289T mutation in the VP1 protein significantly reduces the binding affinity between VP1 and vimentin [55]. This weakened interaction leads to a marked decrease in the virus’s ability to infect the central nervous system. Although vimentin knockout mice provide a vimentin-deficient background, genetic knockout may induce compensatory developmental or physiological changes that could influence viral infection pathways or immune responses. Consequently, this in vivo system only partially models human central nervous system infection and may not fully capture the neuropathological features of *EV71* disease.

### 3.5. Herpesviruses

Recent findings have identified CSV as a critical receptor for *pseudorabies virus (PRV),* a member of the genus *Varicellovirus* and subfamily *Alphaherpesvirinae* of the *Herpesviridae* family [56]. Viral adsorption correlates strongly with vimentin abundance across cell types. Altering vimentin levels through siRNA knockdown or overexpression directly impacts the efficiency of PRV attachment and entry, particularly in HEK-293 cells, which appear more dependent on vimentin than PK-15 cells. This suggests that *PRV* may adopt a cell-type—specific receptor hierarchy in which vimentin can act as a dominant entry factor in some contexts but use additional receptors for compensation when vimentin is less available. Such differential dependence on vimentin between HEK-293 and PK-15 cells also raises the possibility that vimentin functions as a cofactor rather than a universal receptor, and that *PRV* may switch receptor usage depending on host species or tissue context. Mechanistically, this interaction is mediated by the Rod domain of vimentin (96–404 aa), which forms an interface corresponding to a specific hydrophobic patch or acidic groove identified in the models (Figure 2 and Figure 3). On the viral side, key residues in the *PRV* glycoproteins gD (W198, G162, Y164, and C205) and gH (C439) are essential for binding. Structural modeling of gD and gH revealed conserved binding pockets typical of herpesviruses, reinforcing the proposed interaction between *PRV* and vimentin. However, as in the cases of *SARS-CoV-2* and *DENV*, these predicted interaction surfaces remain speculative because they have yet to be confirmed through biophysically validated or high-resolution structure approaches.

## 4. Dual Roles of Vimentin Rearrangement in Viral Replication: Shield or Sabotage

During late infection, the cage-like vimentin structure that forms around the viral VRCs exhibits a context-dependent duality, functioning as either a pro-viral “shield” or an anti-viral “sabotage” strategy (Table 2). Although viruses can exploit this cage’s innate cytoprotective function to localize phase separation processes that compartmentalizes and enhance viral replication, the host can, in other contexts, co-opt vimentin remodeling to mount a successful cellular anti-virus response.

### 4.1. Large Cytoplasmic DNA Viruses

The VRCs generated by large cytoplasmic DNA viruses, such as the *vaccinia virus*, *iridoviruses*, and *African swine fever virus (ASFV)*, contain viral DNA and structural proteins concentrated within the inclusions that serve as sites of replication [75]. Rearranged vimentin forms a cage-like structure around VRCs and is associated with the recruitment of chaperones, proteasomes, and mitochondria, effectively creating a favorable environment for viral replication [16]. To demonstrate that intracellular vimentin facilitates replication, the replication slows in the absence of intracellular vimentin. Vimentin is recruited to viral replication sites during the early stages of *ASFV* infection via a two-step process involving microtubule-mediated transport and phosphorylation [76]. Initially, vimentin filaments are recruited to perinuclear replication sites in a microtubule-dependent manner, observing an “aster” structure at the virus assembly site. Following initiation of viral DNA replication, CaMKII is activated and phosphorylates vimentin. Ser82 phosphorylation weakens interfilamentous interactions, facilitating the redistribution and reorganization of IFs. The phosphorylation level of Ser82 vimentin progressively increases after infection, which is correlated with viral DNA replication and late gene expression. Inhibition of CaMKII activity impedes vimentin phosphorylation and cage formation, ultimately slowing viral DNA replication, underscoring the pivotal role of kinases in this remodeling process. While CaMKII inhibition attenuates viral replication, this intervention alone cannot rule out the involvement of other pathways regulating vimentin dynamics.

### 4.2. Flaviviruses

Some flaviviruses, such as *JEV*, *DENV*, and *ZIKV*, exploit distinct kinase signaling pathways to hijack vimentin. They converge on a common PTM-driven remodeling process: phosphorylation-induced disassembly into oligomers, followed by dephosphorylation-dependent assembly into cage-like structures to support viral replication. However, vimentin rearrangements do not always facilitate replication (Table 2). *Duck Tembusu virus (DTMUV)* appears to be suppressed by intracellular vimentin, possibly as part of a cellular strategy for activating the host’s anti-viral defense.

For *JEV*, the NS1 and NS1’ proteins interact with intracellular vimentin and actively induce its upregulated and rearrangement [77,78]. This process has been thoroughly studied, revealing that NS1 proteins trigger vimentin reorganization through the CDK1-PLK1 signaling axis, which promotes viral replication, thereby highlighting vimentin and its associated pathways as potential anti-viral targets [77]. This establishes a direct link between vimentin cage formation and a specific signaling pathway, rather than considering it a passive or secondary cellular response to infection. NS1 promotes the translocation of CDK1 from the nucleus via Cyclin B-NES/CRM1 [88]. In the cytoplasm, vimentin filament anchors CDK1 and is phosphorylated at Ser56 by CDK1. This modification facilitates the recruitment of PLK1to Ser56 phosphorylated vimentin and then activates it, which subsequently phosphorylates vimentin at Ser83 (Figure 4a). However, the mechanism by which NS1 stimulates CDK1 activity, and how CDK1-dependent anchoring sites on vimentin are generated and recruited to VRCs, and the molecular events connecting these processes remain largely unresolved. These critical steps are described phenomenologically but lack a detailed mechanistic explanation.

Vimentin plays a key structural role in anchoring VRCs to the perinuclear region via direct interactions with NS4A and NS1, as in *DENV*. Teo and Chu [79] observed dynamic changes in NS4A-vimentin colocalization during *DENV2* infection, with colocalization increasing as the infection progressed and peaking at 48 h postinfection. Importantly, vimentin reorganization was not a consequence of early apoptotic events. Instead, NS4A may indirectly induce vimentin rearrangement by activating CaMKII, which phosphorylates vimentin at Ser38 and triggers its reorganization at the perinuclear site (Figure 4b). Additionally, vimentin expression was increased 72h postinfection, further illustrating the vimentin cage. Lei et al. [80] identified the ROCK (Rho-associated kinase) pathway as a key regulator of vimentin dynamics during *DENV* infection, showing that the NS1 protein may contribute to this process by promoting the ROCK-mediated phosphorylation of vimentin at Ser71 that drives its disassembly and reorganization, which is essential for viral replication (Figure 4b). In contrast, during *ZIKV* infection, vimentin interacts with endoplasmic reticulum (ER)-resident RNA-binding proteins, such as RRBP1, suggesting a role in regulating the host RNA-binding environment to support viral replication. Notably, vimentin cage assembly during *ZIKV* infection does not involve phosphorylation at Ser39, Ser56, or Ser83, indicating a phosphorylation-independent mechanism of vimentin remodeling [81]. Instead, *ZIKV* may hijack host proteins such as RRBP1 to rearrange vimentin filaments via alternative PTMs, including tyrosine phosphorylation, acetylation, or proteolytic cleavage. These modifications can influence the solubility, interaction affinity, and subcellular localization of vimentin.

However, emerging evidence reveals that vimentin’s function is neither uniform nor unidirectional across virus species, for *DTMUV*—an enveloped, positive-sense, single-stranded RNA virus of the genus *Flavivirus* (family *Flaviviridae*)—it acts as an anti-viral function. During infection, vimentin colocalizes with the NS1 protein and rearranges into a cage-like defense structure; vimentin overexpression inhibits viral replication, while its silencing promotes the replication, indicating a dual regulatory role [84]. This divergence from the canonical “vimentin-as-pro-viral-scaffold” model raises important conceptual questions about how viruses differentially exploit or counteract cytoskeletal remodeling. One mechanistic clue emerges from the phosphorylation dynamics of vimentin. In BHK-21 cells, phosphorylation of vimentin at Ser56 is upregulated by cyclin-dependent kinase 5 (CDK5), peaks at 24h postinfection, and subsequently decreases after 36h. The temporal pattern suggests that vimentin modification is not merely a structural remodeling event but may represent an early host defensive response. Consistent with this idea, inhibiting CDK5 with Roscovitine disrupts vimentin rearrangement and increases viral replication, confirming that CDK5-dependent phosphorylation of vimentin plays a restrictive rather than supportive role. (Figure 4c). These findings align with broader evidence that IFs form a scaffold complex with CDK5 and modulate its pro-apoptotic activity during oxidant-induced cell death [8,89]. Thus, we hypothesize that DTMUV infection triggers apoptosis-linked kinase activity to induce vimentin fragmentation and promote apoptosis, positioning vimentin as a molecular switch that coordinates structural defense and cell fate decisions (Figure 4c).

### 4.3. Pestivirus

*Classical swine fever virus (CSFV)* is an enveloped RNA virus in the genus *Pestivirus* within the family *Flaviviridae*. Intracellular vimentin forms cage-like structures within the ER, encircling double-stranded RNA during *CSFV* infection, suggesting its pro-viral role in VRC formation [82]. Both the knockdown and overexpression of vimentin significantly affect *CSFV* replication, underscoring the sensitivity of *CSFV* replication to the integrity of the vimentin network. Mechanistically, the phosphorylation of vimentin at Ser72, which is mediated by the RhoA/ROCK signaling pathway, triggers vimentin filament rearrangement. RhoA depletion markedly diminishes ROCK activity and vimentin Ser72 phosphorylation, supporting the existence of a hierarchical RhoA/ROCK-vimentin signaling axis during infection. Notably, vimentin interplays with the viral protein NS5A through its rod domain (96–407 aa), which maps to the predicted high-lipophilicity and electrostatically charged regions (Figure 2 and Figure 3), and this interaction recruits NS5A into a vimentin-based cage-like VRC, ultimately enhancing viral replication. Disruption of vimentin rearrangement inhibited NS5A localization in the ER, implying that vimentin rearrangement is not merely necessary for VRC formation but functionally required for VRC biogenesis. Collectively, these findings highlight a conserved strategy among diverse viruses that manipulate vimentin via host kinase pathways. However, the precise molecular architecture of the NS5A-vimentin interface and how vimentin remodleing integrates with other ER-shaping factors remains insufficiently resolved, highlighting an important avenue for further structural and biophysical investigation.

### 4.4. Picornaviruses

In the context of *EV71* infection, vimentin undergoes pronounced reorganization, forming perinuclear cage-like structures that co-localize with viral particles and newly synthesized viral RNA [83]. Co-immunoprecipitation analyses indicated that vimentin interacts specifically with the viral structural protein VP1 and the protease 3C, suggesting that multiple viral components coordinate cytoskeletal remodeling. To dissect how these interactions alter vimentin dynamics, seven potential phosphorylation sites on the vimentin were screened, revealing that *EV71* selectively induces phosphorylation at Ser82. This modification increases progressively during infection, paralleling the accumulation of VRCs. Crucially, VP1 activates CaMKII, which phosphorylates the N-terminal domain of vimentin at Ser82. Inhibiting CaMKII with KN93 blocks Ser82 phosphorylation, prevents vimentin cage formation, and severely diminishes viral replication, demonstrating that CaMKII-dependent remodeling of vimentin is indispensable for VRC assembly. Interestingly, levels of PKC and Rho-kinase remain unchanged during infection, contrasting with *CSFV*, which relies heavily on the RhoA/ROCK pathway for vimentin rearrangement [82]. This divergence suggests that different virus families converge on vimentin remodeling through distinct kinase pathways, yet the deeper mechanistic logic remains unclear. It is also not yet resolved how VP1activates CaMKII, whether specific scaffolds facilitate kinase recruitment, or how vimentin phosphorylation structurally promotes VRC maturation.

A more complex picture emerges in *foot-and-mouth disease virus (FMDV)*, where vimentin can exhibit pro-viral and anti-viral effects. *FMDV* is a non-enveloped, positive-sense, single-stranded RNA virus belonging to the genus *Aphthovirus* within the family *Picornaviridae*. Similar to the pro-viral role observed in *EV71*, vimentin forms cage-like structures with *FMDV* nonstructural proteins 2C and 3A during infection [85,86]. Vimentin degradation accompanies *FMDV* infection in MCF-10A cells, suggesting that proteolysis facilitates vimentin cage assembly around the viral 2C protein [85]. However, vimentin overexpression significantly suppresses *FMDV* replication, whereas vimentin knockdown markedly enhances the replication, indicating a negative regulatory role of vimentin during *FMDV* replication [86]. As with *DTMUV*, the mechanisms underlying the anti-viral role of vimentin oligomerization during *FMDV* replication remain unclear. We propose a testable hypothesis for this dual regulatory pattern (Figure 4d) that *FMDV* 2C activates caspase 3/7 in BHK-21 cells, resulting in caspase-mediated proteolysis of vimentin, which generates fragments of distinct sizes that are recruited to the viral assembly sites to form a vimentin cage [90,91]. Conversely, the accumulation of vimentin oligomers with exposed amino-terminal proteolytic ends activates caspases to amplify apoptosis, a natural process that eliminates damaged or infected cells, including those harboring viruses [35,92]. A similar vimentin-mediated anti-viral effect has been reported for *porcine circovirus type 2 (PCV2)*. In PK-15 cells, vimentin overexpression enhances *PCV2*-induced caspase-3 activation, and caspase-3 potentially promotes virus-induced apoptosis to suppress *PCV2* replication [87]. However, the study does not define how vimentin mechanistically contributes to apoptosis or caspase-3 activation, as upstream apoptotic triggers were not systematically investigated.

## 5. Targeting Vimentin for Anti-Viral Therapy

Vimentin has emerged as a promising target for anti-viral strategies, with a growing interest in vimentin-targeting agents and recombinant proteins. Administration of the humanized anti-extracellular vimentin (eVIM) monoclonal antibody, hzVSF-v13, to *SARS-CoV-2*-infected Roborovski SH101 hamsters significantly reduced eVIM levels, thereby suppressing inflammation and viral replication [60]. Notably, hzVSF-v13 demonstrated superior therapeutic efficacy compared with that of Remdesivir, highlighting its potential as a broad-spectrum anti-viral agent. Additionally, combination therapy with hzVSF and Tenofovir effectively inhibited woodchuck hepatitis virus infection in vivo [93]. Another promising candidate, the oral compound ALD-R491, exhibits host-directed anti-viral and anti-inflammatory effects, addressing multiple aspects of COVID-19 and related illnesses [53]. A vimentin–Fc fusion protein has also shown efficacy in suppressing *JEV* replication by binding to the virus and preventing its attachment to host cells, which further enhances immune clearance by engaging phagocytes and dendritic cells [94].

However, translating these strategies into clinical applications requires comprehensive in vivo safety and effective evaluations. Despite promising results in animal models, agents such as hzVSF, ALD-R491, and fusion proteins must still be validated in diverse preclinical models and human trials to confirm their safety, pharmacokinetics, and therapeutic efficacy. Additional challenges include the use of the prokaryotic expression system—such as *E. coli*—for producing fusion proteins, which may lack essential mammalian PTMs and thus compromise functionality [95]. Given vimentin’s fundamental roles in cellular plasticity, organelles anchoring, and as a central mediator of cell activation. Future efforts should therefore focus on developing small-molecule inhibitors or antibodies that more precisely disrupt host vimentin oligomer–virus interactions, rather than broadly interfering with vimentin filaments or cage, thereby minimizing off-target effects on fundamental cellular processes.

## 6. Concluding Remarks and Perspectives

Vimentin forms a dynamic, cage-like filamentous network around the nucleus through a multistep assembly process involving coiled-coil dimers, different tetrameric conformations, and bundled protofibrils stabilized by specific domain interactions [15,16,79]. During viral replication, this structural plasticity is exploited to form a protective cage-like structure. Specifically, vimentin depolymerization and polymerization into mature filament networks around VRCs leads to the burial of hydrophobic residues and the exposure of hydrophilic surfaces, thereby likely stabilizing the virus factory and mediating its specific interactions [77]. Despite its importance, the crystallization of the full-length vimentin monomer remains unresolved. High-resolution experimental structures of the complete C-terminal tail (413–466 aa) and N-terminal head (1–86 aa) domains are also still lacking in the PDB. Thus, significant challenges remain in determining the precise crystal structure and understanding the molecular mechanisms underlying these interactions. Our computational analysis provides a mechanistic bridge between the biophysical properties of vimentin and its observed cellular behavior during viral infection. The MLP surface profiling and coulombic electrostatic potential align with the experimental evidence showing that vimentin oligomers support *SARS-CoV-2* attachment and internalization [47,48,49,85], as well as binding interfaces mapped for *PRV* and *CSFV* [56,82]. In addition, phosphorylation and other PTMs such as proteolysis and acetylation are direct drivers of vimentin oligomerization and remain active areas of research [14].

During the virus life cycle, CSVs act as co-receptors or attachment factors that stabilize virus–cell attachment and promote entry, exemplifying a pro-viral “shield” function in viral infection [49,52,57,58,59,64,73,79]. However, the mechanism underlying CSV recruitment remains largely unknown. The vimentin-plectin-β3 integrin recruitment model highlights a key unresolved gap in our understanding of flavivirus infection, a challenge mirrored in *SARS-CoV-2* research where vimentin’s role in assisting ACE2 and the structural elucidation of CSV-S-ACE2 triad are still under active investigation [47,48,49,50,52,53,59]. The vimentin cage represents a conserved structural response with a dual regulatory function in viral replication, providing a protective scaffold that supports viral processes while also modulating anti-viral signaling, including the activation of programmed cell death pathways [96]. Crucially, the “sabotage” process is not mediated by the vimentin cage itself, but rather the cell’s apoptotic response, which involves vimentin fragmentation (via caspases) or pro-apoptotic signaling (via CDK5), both of which are anti-viral responses. We therefore hypothesize that functional dichotomy is determined by distinct PTMs. The “shield” function is a result of non-destructive, kinase-mediated phosphorylation (e.g., CaMKII, CDK1-PLK1, ROCK) that leads to vimentin reorganization into a VRC scaffold. Conversely, the “sabotage” function represents a distinct host defense pathway, triggered by different viruses (e.g., *DTMUV, FMDV, PCV2*), that utilizes different PTMs (e.g., apoptosis-linked kinase activity or caspase-mediated proteolysis) to induce vimentin fragmentation and promote apoptosis. Consequently, targeting vimentin in its pro-viral context presents a promising avenue for therapeutic development.

## Figures and Tables

**Figure 1 ijms-27-00388-f001:**
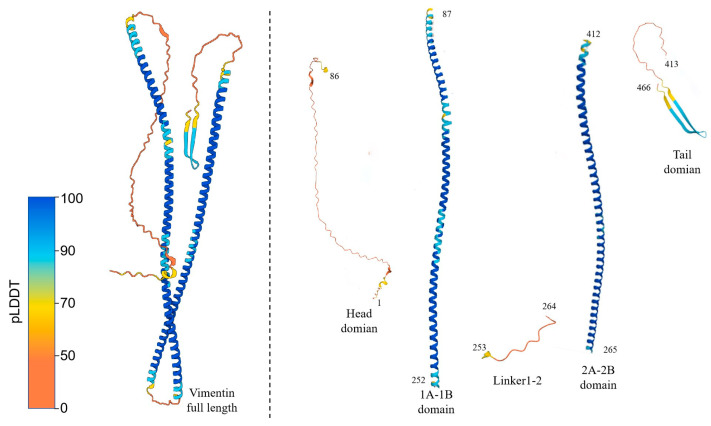
Structural prediction and domain definition of vimentin based on AlphaFold 2 data. The atomic model on the left represents the prediction of the 3D structure of intact vimentin. The residues are color-coded according to the predicted local distance difference test (pLDDT) confidence scores provided by the AlphaFold 2 prediction. On the basis of this model, the vimentin protein domains are defined as follows: the head domain (M1–F86) contains three α-helix fragments; the 1A–1B domain (S87–Q252) and the 2A–2B domain (L265–S412) consist of α-helices connected by Linker 1–2 (H253–D264); and the tail domain (L413–E466) is characterized by two antiparallel β-sheets.

**Figure 2 ijms-27-00388-f002:**
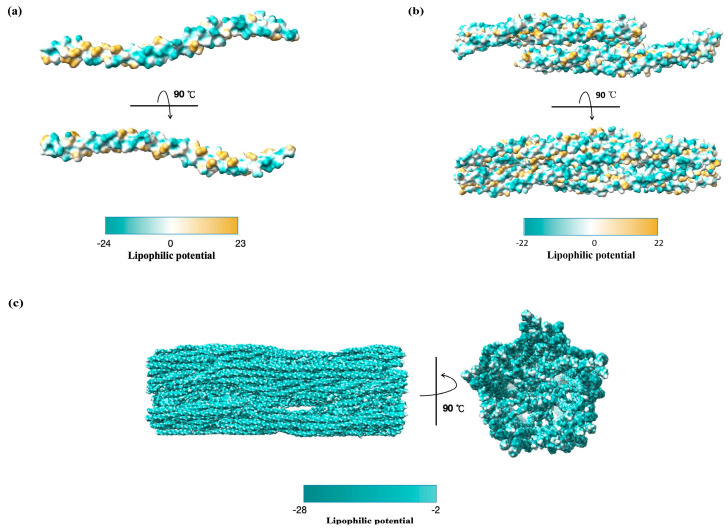
Lipophilic potential of the oligomeric and high-order vimentin surface. (**a**) Orthogonal views of the vimentin monomer (PDB: 4YPC; 161–243 aa) with the molecular surface colored according to lipophilicity, showing hydrophilic and lipophilic residues in dark cyan and gold, respectively. (**b**) Orthogonal views of vimentin octamers (PDB: 5WHF; 153–238 aa), in which the molecular surface exhibits a balanced distribution of hydrophilic and lipophilic areas from top (0°) to bottom (rotated 90°). (**c**) Lipophilic surface potential mapped onto higher-order vimentin structures (PDB: 8RVE; 1–466 aa) reveals an overwhelmingly hydrophilic character on both sides (0° and rotated 90°).

**Figure 3 ijms-27-00388-f003:**
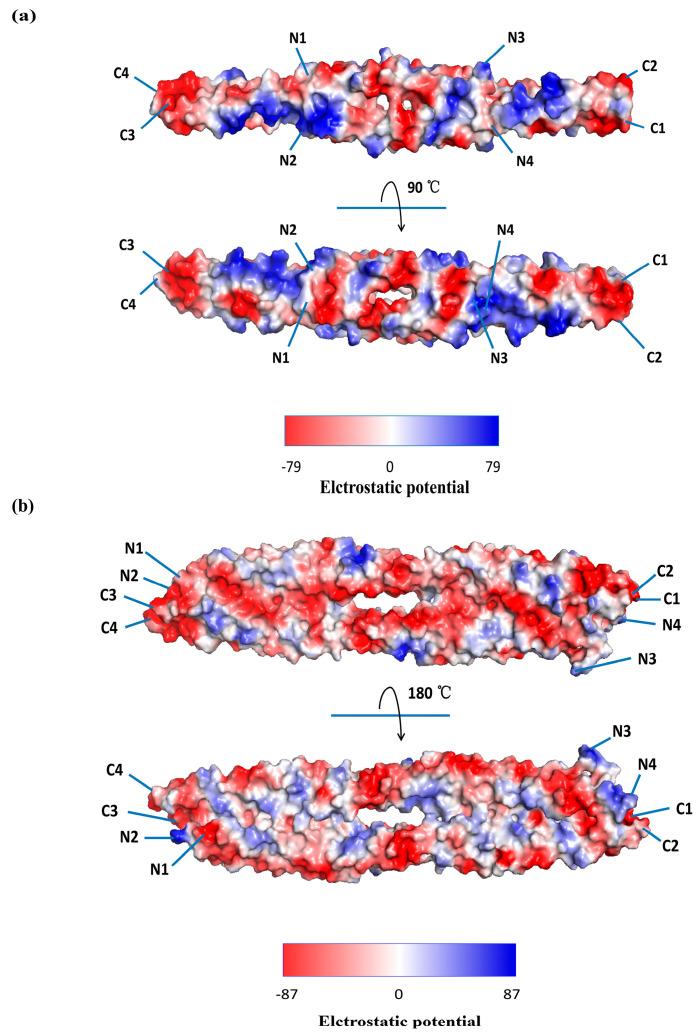
Coulombic electrostatic potential of the tetramers of the vimentin surface. (**a**) The 3KLT (253–324 aa) tetramer surface (top view) displays regions of basic (blue) and acidic (red) charges with two small, highly acidic surface grooves centrally positioned. On the other side (bottom view, rotated 90°), the basic and acidic charges are symmetrically distributed, featuring a prominent, highly acidic surface groove. (**b**) The 3UF1 (144–251 aa) tetramer surface (top view) shows a long, linear, and highly acidic (red) surface groove, whereas the opposite side (bottom view, rotated 180°) features a central concave pocket that is surrounded by alternating patches of negative charges (red) and positive charges (blue). The groove is flanked on both sides by neutral (white) and positively (blue) charged surfaces. N1–N4 and C1–C4 represent the cluster of the four N-termini and C-termini, respectively, from the four vimentin chains within the tetramer. Their mixed arrangement highlights the antiparallel nature of the assembly.

**Figure 4 ijms-27-00388-f004:**
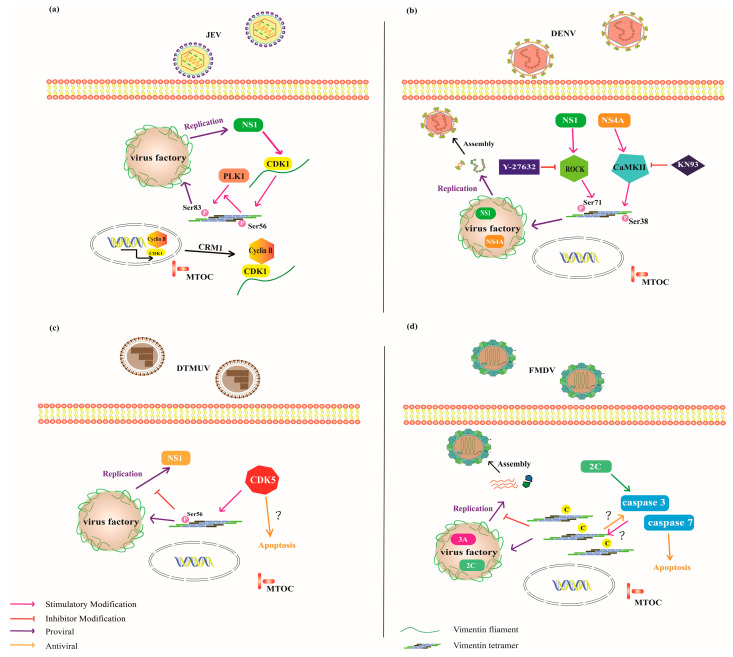
PTM-drive functional switch model explaining the dual pro-viral and anti-viral roles of vimentin during viral replication. (**a**,**b**) In the pro-viral “Shield” model, kinase-mediated phosphorylation (CDK1/PLK1, CaMKII, ROCK), indicated with a magenta arrow, promotes vimentin filament rearrangement into a protective cage that has been associated with enhanced *JEV* and *DENV* replication (purple arrow), potentially by providing a supportive microenvironment for the VRCs. P, phosphorylation; Y27632, inhibitor of ROCK activity; KN93, inhibitor of CaMKII activity. (**c**,**d**) In the anti-viral “Sabotage” model, *FMDV* and *DTMUV* infections are associated with the initial formation of vimentin cage-like structures, followed by a net inhibition of viral replication. We propose that cyclin-dependent kinase 5 (CDK5)-mediated phosphorylation of vimentin (magenta arrow) during *DTMUV* infection contributes to pro-apoptotic responses (orange arrow). In *FMDV* infection, we hypothesize that caspase-mediated cleavage of vimentin and the resulting fragments may facilitate apoptosis (orange arrow), thereby limiting viral replication. P, phosphorylation; C, caspase cleavage.

**Table 1 ijms-27-00388-t001:** Mechanistic insights into CSV binding by various viruses.

Virus Species	Genetic Material	Receptor	Co-Location Viral Protein	Binding Domain on CSV	Cell Line/Animal Model	Model of Interaction	Function	Ref.
*SARS-CoV-2*	*(+)ssRNA*	ACE2	Spike protein	Tail/Rod	A549/Vero E6	Hydrogen bonds/Electrostatic contacts	Synergistic ACE2 combination	[47,48,49,50]
*JEV*	*(+)ssRNA*	α_v_β_3_ integrin	E protein	Head/Tail	BHK-21/HTB-11/N18	Electrostatic contacts	Virus adsorption/endocytosis	[51]
*DENV*	*(+)ssRNA*	DG-SIGN	E(EDIII)	Rod	VECs	Electrostatic contacts	Enhance cell recognition	[52,53]
*HCV*	*(+)ssRNA*	CD81	E1 protein	Head	Huh-7.5.1	-	Cell-to-cell transmission of virus	[54]
*EV71*	*(+)ssRNA*	SCARB2/PSGL-1	VP1	Head	HBMECs/Mice	-	Assist in virus localization	[55]
*PRV*	*(+)dsDNA*	HS	gD/gH	Rod	HEK-293/PK-15	Hydrogen bonds	Virus adsorption	[56]

Note: “-” indicates data not determined or unavailable.

**Table 2 ijms-27-00388-t002:** Mechanistic insights into vimentin rearrangement in viral replication.

Strategy	Virus Species	Co-Location Viral Protein	Signaling Pathway Involved	Vimentin Phosphorylation Sites	Binding Domain on Vimentin	Ref.
Shield (pro-viral function)	Vaccinia virus	p39	-	-	-	[75]
	*Iridoviruses*	-	-	-	-	[75]
	*ASFV*	-	CaMKII	Ser82	-	[76]
	*JEV*	NS1; NS1′	CDK1-PLK1	Ser56; Ser83	-	[77,78]
	*DENV*	NS4A; NS1	CaMKII; ROCK	Ser38; Ser71	-	[79,80]
	*ZIKV*	RRBP1	-	Non-Phosphorylation Ser38; Ser39; Ser56	-	[81]
	*CSFV*	NS5A	RhoA/ROCK	Ser72	Rod	[82]
	*EV71*	VP1; 3C	CaMKII	Ser82		[83]
Sabotage (anti-viral)	*DTMUV*	NS1	CDK5	Ser56	-	[84]
	FMDV	2C; 3A	-	-	-	[85,86]
	*PCV2*	Cap protein	NF-κB; Caspase-3	-	-	[87]

Note: “-” indicates data not determined or unavailable.

## Data Availability

No new data were created or analyzed in this study.

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
