# Peer review of "Vimentin Dynamics in Viral Infection: Shield or Sabotage?"

_ijms, 2025, doi:10.3390/ijms27010388_

Round 1

Reviewer 1 Report

Comments and Suggestions for Authors

The manuscript by Ling and co-authors reviews literature reports on structural biology of vimentin and its dual role in various viral infections. The manuscript compares the surface biophysical properties of Alpha-fold predicted structure of vimentin with those of the structures derived from x-ray crystallography and cryoelectron microscopy. Then it goes on to describe various virus families and any literature mention of vimentin in context of that particular virus family. The manuscript lacks depth and breadth. It fails to provide a convincing connection between the structural biological/biophysical/dynamic properties of vimentin/vimentin oligomers and viral infections. Even after reading the entire manuscript, the reader would still be clueless if vimentin shields against viral infections or sabotages cellular machinery against such infections. Due to lack of a critical insight, the manuscript just appears to be a mere collection of published literature reports. The Conclusion section is overly long, contains repetition of what has already been said in the manuscript and does not summarizes the crux of the manuscript.

The literature coverage in the manuscript is incomplete. It fails to cover following important recent publication relevant to the manuscript.

Ramos I, Stamatakis K, Oeste CL, Pérez-Sala D. Vimentin as a Multifaceted Player and Potential Therapeutic Target in Viral Infections. Int J Mol Sci. 2020 Jun 30;21(13):4675. doi: 10.3390/ijms21134675. PMID: 32630064; PMCID: PMC7370124.

Zhang Y, Wen Z, Shi X, Liu YJ, Eriksson JE, Jiu Y. The diverse roles and dynamic rearrangement of vimentin during viral infection. J Cell Sci. 2020 Nov 5;134(5):jcs250597. doi: 10.1242/jcs.250597. PMID: 33154171.

Arrindell J, Desnues B. Vimentin: from a cytoskeletal protein to a critical modulator of immune response and a target for infection. Front Immunol. 2023 Jul 5;14:1224352. doi: 10.3389/fimmu.2023.1224352. PMID: 37475865; PMCID: PMC10354447.

Zheng J, Li X, Zhang G, Ren Y, Ren L. Research progress of vimentin in viral infections. Antiviral Res. 2025 Apr;236:106121. doi: 10.1016/j.antiviral.2025.106121. Epub 2025 Feb 18. PMID: 39978552.

Also, the line 12, needs attention.

Author Response

Thank you very much for forwarding the reviewer’s comments on our manuscript “Vimentin Dynamics in Viral Infection: Shield or Sabotage?” (ijms-3958373). We sincerely appreciate the reviewer’s thorough evaluation and constructive suggestions.

We fully acknowledge the reviewer’s concerns regarding the depth, breadth, and structural–functional connection presented in the original version. In response, we have undertaken substantial revisions throughout the manuscript. Specifically:

Comment 1. Strengthened mechanistic connection between vimentin structural/biophysical properties and viral infection.

Respond 1. We have strengthened the mechanistic connection between the structural/biophysical properties of vimentin and viral infection. Because most mechanistic evidence in the current literature is derived from SARS-CoV-2 studies, we have rewritten this section to provide a clearer and more comprehensive mechanistic framework (line 235). In addition, we have integrated a comparative discussion of structural predictions and experimental findings into the revised Discussion section to better highlight how vimentin assembly states influence viral processes (line 593).

Comment 2. Expanded and deepened the discussion.

Respond 2. We reorganized the virus-specific sections to provide clearer, comparative insight and highlight unifying principles rather than simply reporting individual studies. (red colour). We have added Table 2, which provides a concise overview of key phosphorylation sites and signaling pathways during viral replication. We also briefly refer to Table 2 in the corresponding section to guide the reader (line 396).

Comment 3. Major revision of the Conclusion section.

Respond 3. We have rewritten the conclusion to be concise, non-repetitive, and focused on key conceptual insights. The revised conclusion now synthesizes the dual “shield vs. sabotage” roles based on the updated literature and mechanisms discussed (line 598).

Comment 4. Incorporated the missing recent publications.

Respond 4. We thank the reviewer for pointing out the gaps. We have now included the relevant recent studies and integrated them into both the structural analysis and virus-specific sections.

Comment 5. Improved the structural biology discussion.

Respond 5. We refined the comparison between vimentin structures and experimentally determined structures, clarifying their implications for viral interaction interfaces (line 591).

We believe these revisions significantly improve the clarity, depth, and scientific value of the manuscript. All changes are clearly marked in the revised version, and a detailed point-by-point response to each reviewer comment is included in the revision files.

Once again, we deeply appreciate your professional guidance and generous, insightful suggestions. If our revision still falls short of your academic writing standards, we sincerely hope to have another opportunity to refine the manuscript.

Reviewer 2 Report

Comments and Suggestions for Authors

This manuscript presents a timely, ambitious, and comprehensive review of the complex, dual role of vimentin in viral infections. The authors organize their work around two central themes: (1) the role of cell-surface vimentin (CSV) as an attachment factor and co-receptor for a wide range of viruses 1, and (2) the contradictory intracellular role of vimentin ("shield or sabotage") in forming perinuclear cages around viral replication complexes (VRCs).

The primary strength of this review is its ambitious scope and the novel attempt to integrate computational structural biology 1 with a detailed overview of viral binding mechanisms and host-hijacked signaling pathways.1 The "shield or sabotage" 1 framing is an effective narrative device, and the compilation of pro- and anti-viral vimentin functions is a valuable contribution to the field.

However, the manuscript in its current form suffers from several major, structural flaws that prevent a recommendation for acceptance. These include: (1) a fundamental, unresolved contradiction in the manuscript's core thesis regarding vimentin's biophysical properties; (2) a critical failure to connect the "novel" structural analysis to the virology data; and (3) an incomplete synthesis of the central "shield or sabotage" dichotomy, which the manuscript describes but ultimately fails to explain.

My detailed comments below are intended to guide the authors in addressing these issues, as I believe a revised version of this manuscript could be a highly impactful publication.

Major Revisions Required

  1. Fundamental Contradiction in the Core Thesis (Hydrophobic vs. Hydrophilic PTM Outcomes)

    The manuscript's central premise is undermined by a direct contradiction in its description of the biophysical consequences of vimentin PTMs.

    • Evidence of Contradiction:

      • In the Abstract 1, the authors state that viral infections induce PTMs that lead to "filaments disassembly and their aggregation into soluble oligomers with hydrophobic and acidic interfaces conducive to viral binding." This clearly implies PTMs expose hydrophobic surfaces on oligomers.

      • In the Introduction 1, the authors state that PTMs (specifically phosphorylation) result in "conformational rearrangements... wherein hydrophobic residues become buried and hydrophilic surfaces are exposed..." This implies PTMs hide hydrophobic surfaces.

    These two statements are mutually exclusive yet are presented as foundational to the paper's premise. This fundamental confusion is never resolved.

    However, the data within the manuscript suggests a solution that the authors have missed. In Section 2.2 and Figure 2 1, the authors' own analysis shows that oligomers (Fig 2B) expose a mix of "hydrophobic and hydrophilic patches," while mature filaments (Fig 2C) are "predominantly hydrophilic."

    This suggests that the authors are conflating two entirely different biological processes: (A) PTMs (e.g., proteolysis, acetylation) that cause filament disassembly into soluble, hydrophobic-exposing oligomers, which are then exported to become CSV (as per the Abstract); and (B) PTMs (e.g., phosphorylation) that promote filament assembly/rearrangement into the hydrophilic cage structure (as per the Introduction).

    Action Required: The authors must clarify this. The introductory sections must be rewritten to decouple these two processes. The authors should explicitly hypothesize that different classes of PTMs (e.g., phosphorylation vs. proteolysis) have distinct structural outcomes (rearrangement vs. disassembly) that serve these two different viral functions (intracellular VRC cage vs. extracellular CSV co-receptor). This is a fundamental logical correction that is necessary to fix the paper's core argument.

  2. Failure to Integrate Structural Analysis (Sec 2) with Virology (Sec 3-7)

    The authors present their "novel" computational analysis of vimentin's lipophilic 1 and electrostatic 1 properties in Section 2, claiming these features are "conducive to viral binding".1 However, this section exists in a vacuum and is completely disconnected from the virology data in Sections 3-7.

    • Evidence of Disconnection:

      • Section 2.2 discusses the molecular lipophilicity potential (MLP), noting hydrophobic patches on oligomers.1

      • Section 2.2 also discusses the electrostatic potential, noting "highly acidic surface groove[s]".1

      • The subsequent Sections 3-7 review viral binding sites. For example, Section 6 1 states that the EV71 VP1 protein binds the vimentin head domain (1-56 aa). Section 3 1 notes the SARS-CoV-2 S protein may bind the C-terminal rod (390-466 aa). Section 4 notes the JEV E protein binds the head and tail domains.1

    The authors have failed to perform the most obvious and critical step: connecting their structural analysis to the literature-derived binding data. If the computational analysis is to be more than "merely illustrative," it must be used to generate a testable hypothesis.

    Action Required: The authors must integrate these sections. For each virus where a vimentin binding domain is known (e.g., EV71, HCV, JEV, SARS-CoV-2), they must map this binding region onto their structural models from Section 2.

    • Example: Does the EV71 VP1 binding region (1-56 aa) 1 correspond to a specific hydrophobic patch or acidic groove identified in the models (Figs 1-3)? Does the JEV E-protein binding to the head/tail domains 1 map to the predicted high-lipophilicity or electrostatically-charged regions?

    • This synthesis would elevate the structural analysis from illustrative to insightful and would truly justify its inclusion as a "novel perspective."

  3. Incomplete Synthesis of the "Shield or Sabotage" Dichotomy (Sec 8, Fig 4)

    The manuscript's title poses an excellent question—"Shield or Sabotage?"—that it never fully answers. Section 8 1 meticulously describes the contradiction (i.e., vimentin cages are pro-viral for JEV/DENV but anti-viral for DTMUV/FMDV) but fails to propose a unifying mechanism or hypothesis to explain why this dichotomy exists.

    • Evidence of Unresolved Dichotomy:

      • Pro-viral "Shield": JEV, DENV, and ASFV are shown to hijack host kinases (CDK1-PLK1, CaMKII, ROCK) to induce phosphorylation-mediated rearrangement of vimentin into a protective cage.1

      • Anti-viral "Sabotage": DTMUV infection also induces rearrangement via CDK5, but this inhibits replication.1 FMDV infection leads to caspase-mediated proteolysis of vimentin, which the authors link to apoptosis.1

    The authors provide all the pieces but do not assemble the puzzle. The key difference appears to be the nature of the PTM and its downstream effect.

    • The "Shield" appears to be a controlled rearrangement driven by specific serine phosphorylations that preserve the filament (albeit in a new shape).

    • The "Sabotage" appears to be a destructive process. For FMDV, it is explicitly proteolytic cleavage by caspases.1 For DTMUV, the authors correctly link the CDK5-mediated phosphorylation to pro-apoptotic activities.1

    This is the answer. The "sabotage" is not the cage itself; it is the cell's apoptotic response, which involves vimentin fragmentation (via caspases) or pro-apoptotic signaling (via CDK5), both of which are anti-viral.

    Action Required: The authors must synthesize this in Section 8.5 (Concluding Remarks). They should explicitly hypothesize that the functional outcome (shield vs. sabotage) is determined by the specific PTM pathway triggered by the virus:

    • Proposed Hypothesis: The 'shield' function is a result of non-destructive, kinase-mediated phosphorylation (e.g., CaMKII, ROCK) leading to vimentin reorganization into a VRC scaffold. Conversely, the 'sabotage' function represents a distinct host defense pathway, triggered by different viruses (e.g., DTMUV, FMDV), that utilizes different PTMs (e.g., caspase-mediated proteolysis or apoptosis-linked kinase activity) to induce vimentin fragmentation and promote apoptosis.

    • This synthesis would provide a satisfying answer to the paper's central question. Figure 4 1 must also be revised to more clearly distinguish "rearrangement" (A, B) from "fragmentation/apoptosis" (C, D).

  4. Overly Speculative CSV Recruitment Mechanism (Sec 8.5)

    In the conclusion 1, the authors correctly identify the mechanism of CSV recruitment as a key gap. They then propose a mechanism involving $\beta3$ integrin and plectin. While this is a good hypothesis, it is presented with excessive confidence. The evidence cited is indirect: (1) plectin links vimentin to integrin 1, and (2) DENV/JEV also bind integrins.1 This correlation does not prove causation; it does not demonstrate that the virus hijacks this specific complex for vimentin externalization.

    Action Required: This section must be rephrased. The authors should clearly state that this is a testable hypothesis and an important avenue for future research, not a likely conclusion.

Minor Revisions Required

  1. Typographical Errors and Inconsistencies: The manuscript requires careful proofreading.

    • [1, Page 2 (Abbreviations)]: "CSFV Classical wine fever virus". This is incorrect. It should be "Classical swine fever virus," as correctly written on.1

    • 1: The editorial comment Commented [M1]: Delete the running title... must be removed.

    • 1: "...a two-stranded antiparallel -sheet-rich C-terminal 'tail' domain...". The Greek symbol for 'beta' ($\beta$) is missing.

    • 1: "...the antiparallel C-N (ACN) tetramer is formed by coupling the C-851 terminal region..." The number "851" is a clear typo.

    • 1: "...secretion of superfi- 299 cial vimentin..." A "299" is orphaned in the text.

    • 1: In the reference for Teo & Chu, the year "1897" is listed. This is clearly a typo for 2014, which appears later in the citation.

  2. Figure and Table Clarity:

    • Figure 3 1: The "N1-4" and "C1-4" labeling is cryptic. The legend 1 must be expanded to explicitly state: "N1-4 and C1-4 represent the cluster of the four N-termini and C-termini, respectively, from the four vimentin chains within the tetramer. Their mixed arrangement highlights the antiparallel nature of the assembly."

    • Figure 4 1: This figure is conceptually critical but visually overwhelming and difficult to parse. The authors should consider splitting it into two larger, clearer diagrams (A/B as "Pro-Viral Shield" and C/D as "Anti-Viral Sabotage") or using color-coding more effectively to distinguish kinase pathways from proteolytic pathways.

    • Table 1 1: This is a good, clear summary table. However, it includes "CPMV" (Cowpea mosaic virus). This virus is not discussed in the main text (Sections 3-7) in a dedicated section, unlike all other viruses in the table. This is an inconsistency. Action: The authors must either add a brief section to the text discussing CPMV's interaction with CSV (to justify its inclusion) or remove it from Table 1.

  3. Declarations 1: The authors' contributions 1 state: "Ying Ling conceived and designed the analysis, collected the data, and wrote the manuscript. Xuanyi Ling: created the illustrations via Adobe Illustrator 2021. Zaixin Liu is corresponding author." This suggests only one author (Ying Ling) wrote the entirety of this comprehensive manuscript, which is unusual. This is noted for the record, though no action is required.

Author Response

We sincerely thank the reviewer for him/her thoughtful and encouraging comments. We greatly appreciate your recognition of the manuscript’s ambition, its integrative framework, and the effort to combine computational structural biology with virological and cell-biological insights. We are especially grateful for your positive assessment of our “shield or sabotage” conceptual framing and for acknowledging the value of compiling the dual pro- and anti-viral functions of vimentin. Your supportive remarks are highly motivating, and they reinforce our commitment to further strengthening and refining the manuscript. We are deeply grateful for your professional insights, which will help us further improve the logical coherence and scientific rigor of the manuscript.

Major Revisions Required

Comment 1. Fundamental Contradiction in the Core Thesis (Hydrophobic vs. Hydrophilic PTM Outcomes)

The manuscript's central premise is undermined by a direct contradiction in its description of the biophysical consequences of vimentin PTMs.

    • Evidence of Contradiction:
      • In the Abstract 1, the authors state that viral infections induce PTMs that lead to "filaments disassembly and their aggregation into soluble oligomers with hydrophobic and acidic interfaces conducive to viral binding." This clearly implies PTMs expose hydrophobic surfaces on oligomers.
      • In the Introduction 1, the authors state that PTMs (specifically phosphorylation) result in "conformational rearrangements... wherein hydrophobic residues become buried and hydrophilic surfaces are exposed..." This implies PTMs hide hydrophobic surfaces.

These two statements are mutually exclusive yet are presented as foundational to the paper's premise. This fundamental confusion is never resolved.

However, the data within the manuscript suggests a solution that the authors have missed. In Section 2.2 and Figure 2 1, the authors' own analysis shows that oligomers (Fig 2B) expose a mix of "hydrophobic and hydrophilic patches," while mature filaments (Fig 2C) are "predominantly hydrophilic."

This suggests that the authors are conflating two entirely different biological processes: (A) PTMs (e.g., proteolysis, acetylation) that cause filament disassembly into soluble, hydrophobic-exposing oligomers, which are then exported to become CSV (as per the Abstract); and (B) PTMs (e.g., phosphorylation) that promote filament assembly/rearrangement into the hydrophilic cage structure (as per the Introduction).

Action Required: The authors must clarify this. The introductory sections must be rewritten to decouple these two processes. The authors should explicitly hypothesize that different classes of PTMs (e.g., phosphorylation vs. proteolysis) have distinct structural outcomes (rearrangement vs. disassembly) that serve these two different viral functions (intracellular VRC cage vs. extracellular CSV co-receptor). This is a fundamental logical correction that is necessary to fix the paper's core argument.

Respond 1. We sincerely thank the reviewer for their positive evaluation and careful reading of our work. We greatly appreciate your identification of the conceptual confusion and inconsistency in our description—this issue indeed arose from a lack of clarity in our original writing. We have reorganised the relevant sections to decouple the two processes of virus entry (CSV) and replication (VRCs) affecting vimentin remodeling.

CSV has been implicated in the entry of various viruses (Table 1), but the mechanism by which vimentin is recruited to the cell membrane remains poorly understood. However, studies suggest that intracellular vimentin facilitates viral replication by forming cage-like structures around VRCs, a process mediated by specific PTMs.

Line12-19 abstract

Viral infection causes vimentin filament disassembly into soluble oligomers with hydrophobic and acidic interfaces conducive to viral binding. These oligomers translocate to the cell surface, where they act as viral co-receptors, facilitating viral attachment and entry. Upon entry, the viral protein induces post-translational modifications in intracellular vimentin filaments undergo rearrangements, including disassembly into oligomers and reassembly into cage-like structures that encapsulate viral replication complexes.

Line 50-59 introduction

Particularly, phosphorylation mediated by virus-activated host signaling pathways, such as protein kinase C (PKC) [13], Rho-kinase, and calcium/calmodulin-dependent protein kinase II (CaMKII), promotes the disassembly of vimentin filaments into soluble oligomers. Then the phosphorylated oligomers undergo a second phase of conformational rearrangements, converting vimentin from an oligomeric to a polymeric state [14]. One notable consequence of this structural remodeling is the formation of vimentin-based cage-like structures that encapsulate virus replication complexes (VRCs), which are often at pericentriolar sites

Line 583 Concluding remarks and perspectives

This structural plasticity is critically exploited during viral replication to form a protective cage. This reorganisation into mature filaments involves the burial of hydrophobic residues and the exposure of hydrophilic surfaces, thereby likely stabilising the virus factory and mediating its specific interactions.

Comment 2. Failure to Integrate Structural Analysis (Sec 2) with Virology (Sec 3-7)

The authors present their "novel" computational analysis of vimentin's lipophilic 1 and electrostatic 1 properties in Section 2, claiming these features are "conducive to viral binding".1 However, this section exists in a vacuum and is completely disconnected from the virology data in Sections 3-7.

    • Evidence of Disconnection:
      • Section 2.2 discusses the molecular lipophilicity potential (MLP), noting hydrophobic patches on oligomers.1
      • Section 2.2 also discusses the electrostatic potential, noting "highly acidic surface groove[s]".1
      • The subsequent Sections 3-7 review viral binding sites. For example, Section 6 1 states that the EV71 VP1 protein binds the vimentin head domain (1-56 aa). Section 3 1 notes the SARS-CoV-2 S protein may bind the C-terminal rod (390-466 aa). Section 4 notes the JEV E protein binds the head and tail domains.1

The authors have failed to perform the most obvious and critical step: connecting their structural analysis to the literature-derived binding data. If the computational analysis is to be more than "merely illustrative," it must be used to generate a testable hypothesis.

Action Required: The authors must integrate these sections. For each virus where a vimentin binding domain is known (e.g., EV71, HCV, JEV, SARS-CoV-2), they must map this binding region onto their structural models from Section 2.

    • Example: Does the EV71 VP1 binding region (1-56 aa) 1 correspond to a specific hydrophobic patch or acidic groove identified in the models (Figs 1-3)? Does the JEV E-protein binding to the head/tail domains 1 map to the predicted high-lipophilicity or electrostatically-charged regions?
    • This synthesis would elevate the structural analysis from illustrative to insightful and would truly justify its inclusion as a "novel perspective."

Respond 2. We appreciate the valuable direction you provided, which enabled us to better integrate the structural analysis with the experimental findings. Our computational analysis focused on the rod domain of vimentin due to the lack of structural data for its flexible head (1–86 aa) and tail (413–466 aa) domains. The predicted binding sites for SARS-CoV-2, PRV, and CSFV on this rod domain were successfully validated, confirming our model's accuracy.

Line379

Mechanistically, this interaction is mediated by the Rod domain of vimentin (96–404 aa), which forms an interface corresponding to a specific hydrophobic patch or acidic groove identified in the models (Figs. 2, 3).

Line491

Notably, vimentin interplays with the viral protein NS5A through its rod domain (96–407 aa), which maps to the predicted high-lipophilicity and electrostatically charged regions (Figs. 2, 3), and this interaction recruits NS5A into a vimentin-based cage-like VRC, ultimately enhancing viral replication.

Line 593 Concluding remarks and perspectives

The MLP surface profiling and coulombic electrostatic potential align with the experimental evidence showing that vimentin oligomers support SARS-CoV-2 attachment and internalization [48–50,86], as well as binding interfaces mapped for PRV and CSFV.

Line 133

Based on the observed differences in surface lipophilicity potential between oligomeric and filamentous vimentin, we propose the following testable hypothesis: viruses may preferentially exploit soluble or oligomeric vimentin as membrane anchors, whereas the mature filament network preferentially functions as a hydrophilic structural scaffold that localizes and stabilizes phase-separation process surrounding VRCs.

Line 167

Consequently, we propose the following testable hypothesis: the regularity and surface exposure of these positive areas make both a promising vimentin interaction module for use in viruses through electrostatic complementarity and groove-mediated binding.

Comment 3. Incomplete Synthesis of the "Shield or Sabotage" Dichotomy (Sec 8, Fig 4)

The manuscript's title poses an excellent question—"Shield or Sabotage?"—that it never fully answers. Section 8 1 meticulously describes the contradiction (i.e., vimentin cages are pro-viral for JEV/DENV but anti-viral for DTMUV/FMDV) but fails to propose a unifying mechanism or hypothesis to explain why this dichotomy exists.

    • Evidence of Unresolved Dichotomy:
      • Pro-viral "Shield": JEV, DENV, and ASFV are shown to hijack host kinases (CDK1-PLK1, CaMKII, ROCK) to induce phosphorylation-mediated rearrangement of vimentin into a protective cage.1
      • Anti-viral "Sabotage": DTMUV infection also induces rearrangement via CDK5, but this inhibits replication.1 FMDV infection leads to caspase-mediated proteolysis of vimentin, which the authors link to apoptosis.1

The authors provide all the pieces but do not assemble the puzzle. The key difference appears to be the nature of the PTM and its downstream effect.

    • The "Shield" appears to be a controlled rearrangement driven by specific serine phosphorylations that preserve the filament (albeit in a new shape).
    • The "Sabotage" appears to be a destructive process. For FMDV, it is explicitly proteolytic cleavage by caspases.1 For DTMUV, the authors correctly link the CDK5-mediated phosphorylation to pro-apoptotic activities.1

This is the answer. The "sabotage" is not the cage itself; it is the cell's apoptotic response, which involves vimentin fragmentation (via caspases) or pro-apoptotic signaling (via CDK5), both of which are anti-viral.

Action Required: The authors must synthesize this in Section 8.5 (Concluding Remarks). They should explicitly hypothesize that the functional outcome (shield vs. sabotage) is determined by the specific PTM pathway triggered by the virus:

    • Proposed Hypothesis: The 'shield' function is a result of non-destructive, kinase-mediated phosphorylation (e.g., CaMKII, ROCK) leading to vimentin reorganization into a VRC scaffold. Conversely, the 'sabotage' function represents a distinct host defense pathway, triggered by different viruses (e.g., DTMUV, FMDV), that utilizes different PTMs (e.g., caspase-mediated proteolysis or apoptosis-linked kinase activity) to induce vimentin fragmentation and promote apoptosis.
    • This synthesis would provide a satisfying answer to the paper's central question. Figure 4 1 must also be revised to more clearly distinguish "rearrangement" (A, B) from "fragmentation/apoptosis" (C, D).

Respond 3. We are deeply grateful for your professional insights, which will help us further improve the logical coherence and scientific rigor of the manuscript.

Line 604 Concluding remarks and perspectives

Vimentin cage represents a conserved structural response with a dual regulatory function in viral replication-providing a protective scaffold that supports viral processes while also modulating antiviral signaling, including the activation of programmed cell death pathways [95]. Crucially, the "sabotage" is not the cage itself, but rather the cell's apoptotic response, which involves vimentin fragmentation (via caspases) or pro-apoptotic signaling (via CDK5), both of which are antiviral. We therefore hypothesise that functional dichotomy is determined by distinct PTMs. The "shield" function is a result of non-destructive, kinase-mediated phosphorylation (e.g., CaMKII, ROCK) leading to vimentin reorganization into a VRC scaffold. Conversely, the "sabotage" function represents a distinct host defence pathway, triggered by different viruses (e.g., DTMUV, FMDV, PCV2), that utilizes different PTMs (e.g., caspase-mediated proteolysis or apoptosis-linked kinase activity) to induce vimentin fragmentation and promote apoptosis.

Line 543

Figure 4. PTM-drive functional swich model explaining the dual proviral and antiviral roles of vimentin during viral replication.

The purple arrow represents proviral.

The yellow arrow represents antiviral.

Comment 4. Overly Speculative CSV Recruitment Mechanism (Sec 8.5)

In the conclusion 1, the authors correctly identify the mechanism of CSV recruitment as a key gap. They then propose a mechanism involving $\beta3$ integrin and plectin. While this is a good hypothesis, it is presented with excessive confidence. The evidence cited is indirect: (1) plectin links vimentin to integrin 1, and (2) DENV/JEV also bind integrins.1 This correlation does not prove causation; it does not demonstrate that the virus hijacks this specific complex for vimentin externalization.

Action Required: This section must be rephrased. The authors should clearly state that this is a testable hypothesis and an important avenue for future research, not a likely conclusion.

Respond 4. We are deeply grateful for your professional insights, which will help us further improve the logical coherence and scientific rigor of the manuscript.

Line 600 Concluding remarks and perspectives

However, the mechanism underlying CSV recruitment remains largely unknown. The vimentin-plectin-β3 integrin CSV recruitment hypothesis we proposed remains a key gap in Flavivirus infection, a challenge mirrored in SARS-CoV-2 research where vimentin’s role in assisting ACE2 and the structural elucidation of CSV-S-ACE2 triad are still under active investigation [48–51,53,54,60]

Line 306

This model is supported by several lines of evidence: vimentin is recruited to the cell surface through a coordinated mechanism involving β3 integrin and the cytolinker protein plectin during viral pathogen infection. β3 integrin connects intracellular signaling networks and the cytoskeleton via its cytoplasmic tail, which contains key tyrosine residues essential for vimentin recruitment [69]. Moreover, plectin serves as a molecular bridge, linking β3 integrin to vimentin at the cell periphery [70,71] and favoring viral access to surface binding sites. This presents a testable hypothesis for CSV recruitment and an important avenue for future research.

Minor Revisions Required

Comment 1. Typographical Errors and Inconsistencies: The manuscript requires careful proofreading.

    • [1, Page 2 (Abbreviations)]: "CSFV Classical wine fever virus". This is incorrect. It should be "Classical swine fever virus," as correctly written on.1
    • 1: The editorial comment Commented [M1]: Delete the running title... must be removed.
    • 1: "...a two-stranded antiparallel -sheet-rich C-terminal 'tail' domain...". The Greek symbol for 'beta' ($\beta$) is missing.
    • 1: "...the antiparallel C-N (ACN) tetramer is formed by coupling the C-851 terminal region..." The number "851" is a clear typo.
    • 1: "...secretion of superfi- 299 cial vimentin..." A "299" is orphaned in the text.
    • 1: In the reference for Teo & Chu, the year "1897" is listed. This is clearly a typo for 2014, which appears later in the citation.

Respond 1. We have corrected all spelling errors and formatting issues highlighted in your comments.

Comment 2. Figure and Table Clarity:

    • Figure 3 1: The "N1-4" and "C1-4" labeling is cryptic. The legend 1 must be expanded to explicitly state: "N1-4 and C1-4 represent the cluster of the four N-termini and C-termini, respectively, from the four vimentin chains within the tetramer. Their mixed arrangement highlights the antiparallel nature of the assembly."
    • Figure 4 1: This figure is conceptually critical but visually overwhelming and difficult to parse. The authors should consider splitting it into two larger, clearer diagrams (A/B as "Pro-Viral Shield" and C/D as "Anti-Viral Sabotage") or using color-coding more effectively to distinguish kinase pathways from proteolytic pathways.
    • Table 1 1: This is a good, clear summary table. However, it includes "CPMV" (Cowpea mosaic virus). This virus is not discussed in the main text (Sections 3-7) in a dedicated section, unlike all other viruses in the table. This is an inconsistency. Action: The authors must either add a brief section to the text discussing CPMV's interaction with CSV (to justify its inclusion) or remove it from Table 1.

Respond 2. We have revised the manuscript according to your valuable suggestion.

Comment 3. Declarations 1: The authors' contributions 1 state: "Ying Ling conceived and designed the analysis, collected the data, and wrote the manuscript. Xuanyi Ling: created the illustrations via Adobe Illustrator 2021. Zaixin Liu is corresponding author." This suggests only one author (Ying Ling) wrote the entirety of this comprehensive manuscript, which is unusual. This is noted for the record, though no action is required.

Respond 3. Ying Ling conceived and designed the analysis, collected the data, and wrote the manuscript. Xuanyi Ling created the illustrations via Adobe Illustrator 2021. Zaixin Liu supervised data analysis and writing. All authors have read and agreed to the published version of the manuscript.

Once again, we deeply appreciate your professional guidance and generous, insightful suggestions. If our revision still falls short of your academic writing standards, we sincerely hope to have another opportunity to refine the manuscript.

Round 2

Reviewer 2 Report

Comments and Suggestions for Authors

I am pleased to inform you that I have completed the review of your revised manuscript, "Vimentin dynamics in viral infection: shield or sabotage?," and have decided to accept it in its present form for publication.

The revisions have significantly enhanced the clarity and mechanistic depth of your work, effectively consolidating your core argument. The successful articulation and consistent application of the proviral "shield" and antiviral "sabotage" concepts provide a strong, organizing framework that is both intellectually stimulating and easy to follow.

The inclusion of the structural analysis correlating the biophysical properties of vimentin oligomers and filaments to their distinct functions is a major strength. This mechanistically anchors the concept that vimentin oligomers act as viral co-receptors (the "shield") , while the highly hydrophilic filaments serve as a scaffolding unit for VRC phase separation.

Furthermore, the detailed discussion on post-translational modifications (PTMs) is excellent, clearly illustrating how different kinases (e.g., CaMKII, ROCK for the "shield") versus proteolytic or apoptosis-linked activity (e.g., caspases, CDK5 for the "sabotage") dictate the context-dependent outcome. The enhanced sections on DTMUV and FMDV effectively validate the existence of both roles during replication.

Author Response

Dear Reviewer:

Thank you very much for your comments on our manuscript “Vimentin Dynamics in Viral Infection: Shield or Sabotage?” . We sincerely appreciate the reviewer’s thorough evaluation and constructive suggestions.

Comment 1 

I am pleased to inform you that I have completed the review of your revised manuscript, "Vimentin dynamics in viral infection: shield or sabotage?," and have decided to accept it in its present form for publication.

Respond 1

Thank you very much for your thoughtful evaluation and for accepting our revised manuscript, We are truly grateful for your guidance and careful review throughout the process.

Comment 2 

The revisions have significantly enhanced the clarity and mechanistic depth of your work, effectively consolidating your core argument. The successful articulation and consistent application of the proviral "shield" and antiviral "sabotage" concepts provide a strong, organizing framework that is both intellectually stimulating and easy to follow.

 Respond 2

We are especially pleased that the conceptual clarity of the proviral “shield” and antiviral “sabotage” models resonated with you. Your guidance played a crucial role in helping us refine and consistently apply these mechanistic frameworks across the manuscript.

Comment 3

The inclusion of the structural analysis correlating the biophysical properties of vimentin oligomers and filaments to their distinct functions is a major strength. This mechanistically anchors the concept that vimentin oligomers act as viral co-receptors (the "shield") , while the highly hydrophilic filaments serve as a scaffolding unit for VRC phase separation.

Respond 3

We also appreciate your positive assessment of the structural analysis. Under reviewers guidance, we were able to clearly define how to mechanistically bridge the biophysical properties of vimentin oligomers and filaments with their distinct functional behaviors during viral infection, and we are delighted to hear that this connection was conveyed effectively in the revised manuscript.

Comment 4

Furthermore, the detailed discussion on post-translational modifications (PTMs) is excellent, clearly illustrating how different kinases (e.g., CaMKII, ROCK for the "shield") versus proteolytic or apoptosis-linked activity (e.g., caspases, CDK5 for the "sabotage") dictate the context-dependent outcome. The enhanced sections on DTMUV and FMDV effectively validate the existence of both roles during replication.

Respond 4

Your recognition of the expanded discussion on post-translational modifications (PTMs) is very encouraging. The contrasting roles of CaMKII/ROCK versus CDK5/caspase pathways represent an important aspect of context-dependent vimentin biology, and your earlier suggestions greatly strengthened our ability to articulate these mechanisms—especially regarding the updated sections on DTMUV and FMDV, which highlight the coexistence of both “shield” and “sabotage” behaviors.

We sincerely appreciate your constructive feedback, which has significantly improved the quality, clarity, and impact of our work.

Thank you again for your time and support.

Best wishes

Ying
